# Transmission-Blocking Vaccines for Canine Visceral Leishmaniasis: New Progress and Yet New Challenges

**DOI:** 10.3390/vaccines11101565

**Published:** 2023-10-05

**Authors:** Jaqueline Costa Leite, Ana Alice Maia Gonçalves, Diana Souza de Oliveira, Lucilene Aparecida Resende, Diego Fernandes Vilas Boas, Helen Silva Ribeiro, Diogo Fonseca Soares Pereira, Augusto Ventura da Silva, Reysla Maria da Silveira Mariano, Pedro Campos Carvalhaes Reis, Eiji Nakasone Nakasone, João Carlos França-Silva, Alexsandro Sobreira Galdino, Paulo Ricardo de Oliveira Paes, Marília Martins Melo, Edelberto Santos Dias, Miguel Angel Chávez-Fumagalli, Denise da Silveira-Lemos, Walderez Ornelas Dutra, Rodolfo Cordeiro Giunchetti

**Affiliations:** 1Laboratory of Biology of Cell Interactions, Department of Morphology, Federal University of Minas Gerais (UFMG), Belo Horizonte 31270-901, MG, Brazil; jaquecleite@gmail.com (J.C.L.); anafish@hotmail.com (A.A.M.G.); dianaso@ufmg.br (D.S.d.O.); lucileneresendeo@yahoo.com.br (L.A.R.); diegofervboas@ufmg.br (D.F.V.B.); helenribeiro078@gmail.com (H.S.R.); diogofsp@ufmg.br (D.F.S.P.); augustovent@ufmg.br (A.V.d.S.); reyslamariano@yahoo.com.br (R.M.d.S.M.); pedroccreis160@gmail.com (P.C.C.R.); eijinakasone@ufmg.br (E.N.N.); franca@icb.ufmg.br (J.C.F.-S.); denise.lemos@gmail.com (D.d.S.-L.); waldutra@icb.ufmg.br (W.O.D.); 2Microorganism Biotechnology Laboratory, Federal University of São João Del-Rei (UFSJ), Midwest Campus, Divinópolis 35501-296, MG, Brazil; asgaldino@ufsj.edu.br; 3Department of Veterinary Clinic and Surgery, School of Veterinary, Federal University of Minas Gerais (UFMG), Belo Horizonte 31270-901, MG, Brazil; paulopaes@vet.ufmg.br (P.R.d.O.P.); mariliamartinsmelovet@hotmail.com (M.M.M.); 4René Rachou Research Center, Oswaldo Cruz Foundation, Belo Horizonte 30190-002, MG, Brazil; edelberto.dias@fiocruz.br; 5Computational Biology and Chemistry Research Group, Vicerrectorado de Investigación, Universidad Católica de Santa María, Arequipa 04000, Peru; mchavezf@ucsm.edu.pe

**Keywords:** canine visceral leishmaniasis, vaccines, transmission-blocking vaccines

## Abstract

Dogs with visceral leishmaniasis play a key role in the transmission cycle of *Leishmania infantum* to humans in the urban environment. There is a consensus regarding the importance of developing a vaccine to control this disease. Despite many efforts to develop a protective vaccine against CVL, the ones currently available, Leish-tec^®^ and LetiFend^®^, have limited effectiveness. This is due, in part, to the complexity of the immune response of the naturally infected dogs against the parasite and the complexity of the parasite transmission cycle. Thus, strategies, such as the development of a transmission-blocking vaccines (TBVs) already being applied to other vector-borne diseases like malaria and dengue, would be an attractive alternative to control leishmaniasis. TBVs induce the production of antibodies in the vertebrate host, which can inhibit parasite development in the vector and/or interfere with aspects of vector biology, leading to an interruption of parasite transmission. To date, there are few TBV studies for CVL and other leishmaniasis forms. However, the few studies that exist show promising results, thus justifying the further development of this approach.

## 1. Introduction

Visceral leishmaniasis (VL), the most severe form of leishmaniasis [1,2], is caused by *Leishmania (Leishmania) donovani* [3] in the Old World, and *Leishmania (Leishmania) infantum* [4] in both the Old and New World (syn *L. chagasi*). VL, caused by *L. infantum*, is a zoonosis with dogs (*Canis familiaris*) playing a key role as reservoirs of *L. infantum* protozoan, once the canine host is the main domestic reservoir [5,6,7]. Canines are considered natural reservoirs because the parasites can survive in skin macrophages, facilitating the transmission to vectors [8]. Indeed, a substantial overlap between locations where human cases are detected and high canine seroprevalence have been reported, underscoring the close relationship between canine and human infections [8,9,10,11]. The vector *Lutzomyia longipalpis*, described by Lutz and Neiva (1912), accounts for 90% of VL transmissions in Latin America [12,13]. 

In Latin America, Brazil represents the main VL endemic country [14] and the recommended control measures by the Health Ministry in Brazil include the treatment of patients with the disease, usage of insecticides, and euthanasia of infected dogs [15]. Despite some studies showing that eliminating seropositive dogs reduces the incidence of VL in both dogs and humans [16,17,18], the current scenario shows that the euthanasia of dogs does not solve the problem of parasite transmission [15,19,20,21,22]. In addition to being controversial among researchers, euthanasia of infected dogs has been harshly criticized from an ethical point of view, in addition to not being well accepted by tutors. 

Upon all the limitations that control programs face, vaccination is considered the most cost-effective control tool for human and canine diseases [23,24]. Thus, the development of vaccines against CVL remains a priority. To achieve this goal, the development of new strategies to control parasite transmission is a pressing need. The present review aims to briefly describe the available CVL vaccines as well as their drawbacks, emphasizing the potential of transmission-blocking strategies as innovative tools to ultimately prevent disease.

## 2. Canine Visceral Leishmaniasis Immunology and Commercial Vaccines

The outcome of the CVL is influenced by the parasite species and the host’s immune response [5,25,26]. Despite the complexity of the disease and the different clinical signs that dogs can exhibit, it has already been shown that the resistance profile is associated with a strong induction of a Th1 response, with the production of IL-12, IFN-γ, IL-2 and TNF-α. Conversely, a Th2 profile, including the cytokines IL-4, IL-5, IL-10, IL-13 and TGF-β, is related to susceptibility [26,27,28,29,30,31,32]. The compartmentalized organ-specific immune response has been associated with both granuloma maturation in liver and antileishmanial activity in *L. donovani* infection [33]. 

In *L. infantum* naturally infected dogs, the immunophenotypic profile of peripheral blood cells showed a prominent reduction in the absolute numbers of total CD5^+^ T-cells and their T-cells subsets (CD4^+^ and CD8^+^) [26,34]. Both oligosymptomatic and symptomatic dogs usually have a predominantly Th2 immune response, which are correlated with disease progression and high parasitism [26,32,35,36,37,38]. Notably, a characteristic of disease susceptibility is the proliferation of non-immunoprotective B-cells, due to the depletion of T-cells in dogs with high-parasite loads [39]. An increase in total IgG levels has been observed in oligosymptomatic and symptomatic dogs compared to healthy and asymptomatic dogs [40,41,42]. 

It has currently been established that a vaccine against CVL represents a critical tool to controlling both human and canine cases [23,26], and a better understanding of the immunology behind the resistance and susceptibility profile is essential for the development of vaccines against CVL [43]. Studies focusing on the development of vaccine candidates have increased in recent years and, to date, there are only two commercially available vaccines for CVL: Leish-Tec^®^ (CEVA, Paulínia, SP, Brazil), and LetiFend^®^ (LETIPharma, Barcelona, Spain). However, more recently, after verifying product compliance deviation, which may compromise the vaccine’s effectiveness, the Brazilian Ministry of Agriculture and Livestock determined the suspension of the manufacture and sale of Leish-Tec^®^ and, so far, new recommendations are being awaited [44]. Moreover, after ten years of commercialization, CaniLeish^®^ (Virbac, Carros, France) was withdrawn from the European market in 2021 [45]. 

Leish-Tec^®^ was licensed in 2014 and is the only available vaccine authorized for use in Brazil. It consists of recombinant A2 (rA2) protein, which was the first to be identified as the amastigote-specific virulence factor in *Leishmania* [46], supplemented with the adjuvant saponin. Fernandes and colleagues observed that the vaccinated dogs produced higher IFN-γ levels. However, despite the high IFN- γ production, parasites were detected in 57.14% and 28.5% of the bone marrow and blood of vaccinated animals, respectively. The authors concluded that immunization with rA2 was immunogenic and able to provide partial protection in dogs [47]. Afterward, Regina-Silva and coauthors demonstrated that vaccinated dogs presented higher levels of total IgG, IgG2, and IgG1 anti-A2 when compared to the control group. In addition, analysis of parasitological exams and xenodiagnosis demonstrated an efficacy of 58.1% [46]. Subsequently, Grimaldi et al. showed the same antibody production with the aforementioned study in vaccinated animals in a field trial; however, 26.49% of the dogs converted to a seropositive status and 43% of the vaccinated dogs developed the disease over time. The authors concluded that Leish-Tec^®^ offers promising results, however, it needs to be optimized to ensure efficacy in dogs under field conditions [48]. Finally, Aguiar-Soares and colleagues reaffirmed the vaccine’s ability to induce increased IFN-γ production by CD8^+^ T-cells [49]. 

The vaccine LetiFend^®^ was authorized for use in Europe in 2016 [50]. This formulation consists of the Protein Q, which is a genetic fusion of five antigenic fragments from four *L. infantum* proteins, namely acidic ribosomal proteins Lip2a, Lip2b, LiP0, and the histone H2A, without an adjuvant [50]. Molano and coworkers were the first to test the Protein Q in association with Bacillus Calmette-Guérin (BCG) as adjuvant and demonstrated long-lasting cellular and humoral responses and activation in macrophages to produce NO [51]. Moreover, the vaccine triggered a positive delayed-type hypersensitivity (DTH), and after experimental infection vaccinated dogs showed lower intensity of clinical symptoms [52]. Lastly, Cotrina and coauthors showed that Protein Q induced 72% efficacy in preventing clinical cases of CVL. The authors concluded that LetiFend^®^ is safe and lowers the risk of developing CVL clinical signs [53]. The main characteristics of the commercial vaccines for CVL discussed in this section are summarized in Table 1.

The studies briefly discussed above indicate that the commercially available vaccines have limitations and none of them are 100% effective, meaning that vaccinated dogs could still become infected with *L. infantum*. Some researchers consider their efficacy limited and believe that vaccines may interfere with the interpretation of serological tests for dog disease diagnosis [21]. In addition, there are still few studies that analyze xenodiagnosis and, due to this, it is not known for sure whether these vaccines can impact the transmission of the parasite. However, the progress made so far is believed to be the basis for developing more effective vaccines [54]. Developing an ideal vaccine is far from an easy task due to the parasite’s antigenic and biological complexity and its ability to evade the host’s immune response [55]. The ideal vaccine, in addition to protecting the vaccinated animal, should be capable of interrupting the parasite’s transmission cycle. To circumvent the problems related to the transmission of the parasite, TBV have been gaining ground and have shown promising results in interfering with the biological cycle of vectors [56].

## 3. Development of Transmission-Blocking Vaccines (TBVs): A Strategy to Interrupt Pathogens Transmission

The development of TBVs stands out as an approach that provides collective protection, since it aims to disrupt the pathogen’s transmission chain by vectors [56]. The principle of this type of vaccine is to use vector or parasite antigens to induce the production of antibodies against the vector or against the parasite. These antibodies will likely interfere with the pathogen’s survival or virulence in the vector, and with the biological aspects of the vector. Thus, feeding on a vaccinated and infected host would reduce vector competence parasite transmission [57].

The first attempts to develop TBVs were related to malaria control, aiming to interrupt or reduce the transmission cycle by targeting the reproductive initial stages of malarial parasites [58,59]. Membrane proteins, such as Pvs25 and Pvs28, expressed on the surface of parasite’s zygotes and ookinetes have been extensively studied for *Plasmodium vivax’s* TBV, effectively suppressing the development of ookinetes in mosquitoes [60,61]. The use of polyclonal anti-midgut antibodies blocked the development of both *Plasmodium falciparum* and *Plasmodium vivax* parasites in five different species of mosquitoes, reducing mosquito survival and fecundity. These data revealed the potential use of antibodies for the development of vaccines against such midgut receptors [62].

Other candidates for TBVs, such as the monoclonal antibodies (MG96) binding to the midgut glycoproteins of *Anopheles stephensi*, resulted in a 100% dose-dependent blockade against *Plasmodium yoelii* development in the vector midgut [63]. The glycoprotein from *Anopheles gambiae* aminopeptidase N glycoprotein (AgAPN1), which is a target for Jacalin (lectin), plays an important role in inhibiting ookinete attachment by masking glycan ligands on midgut epithelial surface glycoproteins. The α-AgAPN1 anti-IgG strongly inhibited both *Plasmodium berghei* and *Plasmodium falciparum* development in different mosquito species, implying that the glycoprotein has a conserved role in ookinete invasion of the midgut and, therefore, may be a target for the development of TBVs for malaria control [64]. Another important molecule involved in the establishment of *Plasmodium falciparum* infection in the *Anopheles gambiae* midgut is Carboxypeptidase B (CPB). The addition of antibodies directed against the carboxypeptidase gene (CPBAg1) to a *P. falciparum*-containing blood meal inhibited CPB activity and blocked parasite development in the midgut [65]. Although TBV candidates have shown good results when it comes to controlling the transmission of *Plasmodium sp.* and the survival of the vectors, there are still no studies in humans that attest to the safety and efficacy of this approach in controlling malaria.

In addition to malaria, TBVs have already been tested for other vector-borne diseases. In the 1990s, Ramasamy and coworkers showed that feeding *Aedes aegypti* with blood from vaccinated animals resulted in a reduction in susceptibility to infection by the Ross River virus and Murray Valley encephalitis virus [66]. Similarly, Ramasamy and Ramasamy demonstrated that antibodies generated by mice immunized with *Anopheles farauti* midgut antigens reduce the number of *P. berghei* oocytes developing in the vector [67]. Notably, antigens from *A. aegypti* demonstrated a noteworthy performance against the mosquito cycle by up to 90% [68]. Finally, TBVs have already been described for other diseases and vectors, such as the West Nile virus and ticks [69].

## 4. Development of TBVs for the Control of Leishmaniasis

The use of sandfly antigens to develop vaccines for leishmaniasis control started in 1996. In the first study, hamsters were immunized with different concentrations of sandfly gut antigens. After three doses, *Phlebotomus duboscqi* fed directly on these vaccinated animals. The authors showed an increase in *P. duboscqi*-specific IgG antibodies production in the vaccinated animals, in addition to increased sandfly mortality and reduction in egg production [70]. Later, Tonui and coworkers immunized mice with crude whole parasites, rgp63, LPG or a cocktail containing rgp63 and LPG, all derived from *Leishmania major*. After immunization, these vaccinated animals were infected with *L. major* and then, *P. duboscqi* fed directly on these animals. They observed the lowest infection rates as well as impairment in parasite development [71]. Kamhawi et al. immunized mice with PpGalec, a galectin reported in the midgut of the *Phlebotomus papatasi*. After five immunizations the sandflies were submitted to artificial feeding containing *L. major* parasites and antibodies triggered by immunization. A reduction of *L. major* promastigotes in the midgut of these vectors was observed [72].

Using a different approach, in the study conducted by Vilela and colleagues, rabbits were immunized through repeated sandfly bites and *L. longipalpis* females fed directly on these vaccinated animals. The authors observed a decline in the fecundity of these sandflies, in addition to the increased mortality of females [73]. In that same year, Saraiva et al. conducted a study to evaluate the performance of the Leishmune^®^ vaccine as TBV in dogs. After animals’ vaccination, an artificial sandfly feeding was employed in order to access the infection of *L. longipalpis* females with *L. chagasi*. For this purpose, a chick-skin membrane was used, where sandfly fed of dog’s immunized sera plus *L. chagasi* parasites. The authors observed a decrease of 20.7% in infection index, proved by the ability to block the attachment of parasites in the midgut and by the lower rate of parasite infection [74]. Although Leishmune^®^ did not contain vector antigens in its formulation, this vaccine was able to interfere in *Leishmania* infection in sandflies. However, in 2014, its commercialization was suspended in Brazil for not meeting the requirements of the phase III study [75].

Coutinho-Abreu and coauthors identified a possible target for TBV. In that study, the authors induced knockdown of PpChit1 transcripts through injection of dsRNA into the *P. papatasi* thorax. Mice were immunized with PpChit1 and the serum was collected. After the blood meal, containing *L. major* and the serum of immunized animals, a reduction in promastigotes present in the midgut of infected *P. papatasi* was observed [76]. 

Bongiorno and colleagues evaluated the ability of the vaccine CaniLeish^®^ to act as a TBV. In this study, vaccinated and naturally infected dogs were exposed directly to *Phlebotomus perniciosus*. The sandflies that fed on vaccinated animals had a lower rate of infection, in addition to having a lower parasite load in the midgut [77]. Although this vaccine was not developed as TBV and does not contain vector antigens in its formulation, these results are promising, however, further studies are needed to assess the real ability of CaniLeish^®^ to act as a TBV. The main findings of TBVs discussed in this section are summarized in Table 2.

In addition to the studies mentioned above, several groups have been studying the use of vector antigens incorporated into vaccines, especially for VL, associated or not with parasite antigens. However, these vaccines were intended to evaluate only the protection against the development of infection. Despite several studies demonstrating promising efficacy, these vaccines have not been tested as TBV. Thus, the field of TBV for leishmaniasis is still little explored, despite the promising results obtained in the few existing studies. Moreover, despite the great biotechnological potential of TBVs, to our knowledge, none of the studies cited above resulted in patents deposit. In fact, only patents related to CaniLeish^®^ and Leishmune^®^ were found in the databases. However, these vaccines were not reported as TBVs.

## 5. Conclusions: Future Trends and Perspectives of TBVs for Visceral Leishmaniasis Control

Although commercially available vaccines for use in dogs lead to protective immune responses, there is still an important bottleneck in demonstrating the effectiveness in preventing parasitic transmission to the vector. Therefore, new strategies are required to control the vector and the transmission of the parasite, obtaining an effective reduction in the number of cases of canine and human disease. In this sense, the development of TBVs for dogs using antigens from the vector could be an important strategy for controlling the spread of VL, given that the antibodies produced in the vaccinated dog could prevent parasite development in the vector. Notably, important physiological events occur after sandfly blood meals interfering with the host immune system effector mechanisms such as complement activation [78]. The result is the establishment of *Leishmania* infection in the host, demonstrating its importance for parasite survival. In this context, the induction of specific antibody production in the host could contribute to reduce the parasite’s persistence in the sandfly [56].

Additionally, for achieving the expected effects of TBV, it is first necessary to carry out the rational selection of antigens. In this sense, several studies have used salivary antigens in association with *Leishmania* antigens to evaluate vaccine protection in the vertebrate host, with promising results (Table 2). However, new studies are required to verify their ability to interrupt the parasite transmission, especially in dogs, which have an important role as reservoirs in *L. infantum* transmission. Regardless of the type of antigen used, it is possible to hypothesize that those related to the physiology of sandflies resulting in a successful blood meal could be more promising antigen candidates to compose a TBV formulation. Although the studies available so far have demonstrated the effect on the vector biological cycle and on infection, the field of TBVs lacks studies that demonstrate how antibody–vector interaction occurs and how these antibodies exert their effect. One hypothesis is that antibodies induced by TBV would damage epithelial cells in sandfly midgut. This hypothesis was supported by the observed degeneration of sandfly midgut epithelium, accompanied by a decrease in the *L. major* infection [71]. However, further studies are needed to demonstrate such interactions and their effects at the cellular level.

The selection of parasite antigens to act as TBV should take their biological effect under consideration. In fact, some molecules have been described as having an important role in the parasite–vector interaction, such as lipophosphoglycan (LPG). It is known that LPG is extremely important for the attachment of the parasite to the intestinal epithelium of the vector and therefore they can escape from the peritrophic matrix and thus avoid elimination with the faecal bolus [79,80,81]. However, new studies are required to evaluate how such antibodies could exert their function as TBV. Even though TBVs are different from current commercially available vaccines, they must also fit within certain criteria, such as safety, reproducibility and allow for large-scale and cost-effective production [82]. Furthermore, there are other criteria that TBV candidates must follow, such as (*i*) inducing high-antibody titers in the vaccinated animal so that it can exert the effect on the vector, and (*ii*) display low levels of polymorphisms [58]. Moreover, one of the greatest challenges in developing TBVs is the ability to maintain high-antibody titers in the vaccinated animal, especially due to the nature of the immunizer, since most of the proposed antigens are not normally found in the host [69]. Despite the progress, there are still important points to be clarified: (*i*) if the proteases activated by the blood meal, especially trypsin, can compromise the biological action of the ingested antibody and how this can present an obstacle to the development of TBVs; (*ii*) how long the activity of the antibodies can be maintained in the vector; and (*iii*) what is the minimum time required for the antibody to trigger its biological action in the vector?

Regardless of its promising effects, the field of TBVs in CVL and in leishmaniasis is still little explored. This could be due to (*i*) the difficulty to maintain the vector under laboratory conditions, due to need of proper infrastructure and trained professionals [83]; (*ii*) the complexity of finding an antigen that is capable of inducing high titers of antibodies in the host and being able, at the same time, to interfere with biological aspects of the vector and also in blocking the binding of the parasite to the vector; or even (*iii*) that efforts are more focused on the development of vaccines that protect the host against infection, taking the focus away from the TBV’s strategy. Lastly, most TBV studies were performed under laboratory conditions, which may mask the real effectiveness of the antigens tested. In this sense, it is necessary to identify promising antigens to compose TBVs to act in the control of the transmission of leishmaniasis, in addition to the need to carry out field tests to verify the real effectiveness of this strategy.

Our research group has already identified different *L. longipalpis* antigens that could be used in this promising strategy. These antigens showed the ability to exert the two main goals of a TBV: (*i*) alter the insect’s homeostasis, thus leading to death or reduced oviposition; and/or (*ii*) reduce the parasite load in the vector’s intestine (Figure 1) [32,56,68,84,85]. Although these antigens showed excellent results, more studies are needed to evaluate their performance under field conditions. In addition, these sandfly antigens could be incorporated with new *Leishmania* antigens and/or in association with commercially available vaccines to improve the control of parasite transmission in VL endemic areas. Since the antibodies induced by vaccination prevent parasite development in the insect and its subsequent transmission, interrupting the epidemiological cycle becomes possible, and preventing human and canine VL cases would then be feasible. Another TBV advantage would be to allow the safe treatment of infected dogs without compromising human and other animal health. The field of TBVs should obtain more focus in the coming years, as it presents a promising new strategy to effectively control transmission.

## Figures and Tables

**Figure 1 vaccines-11-01565-f001:**
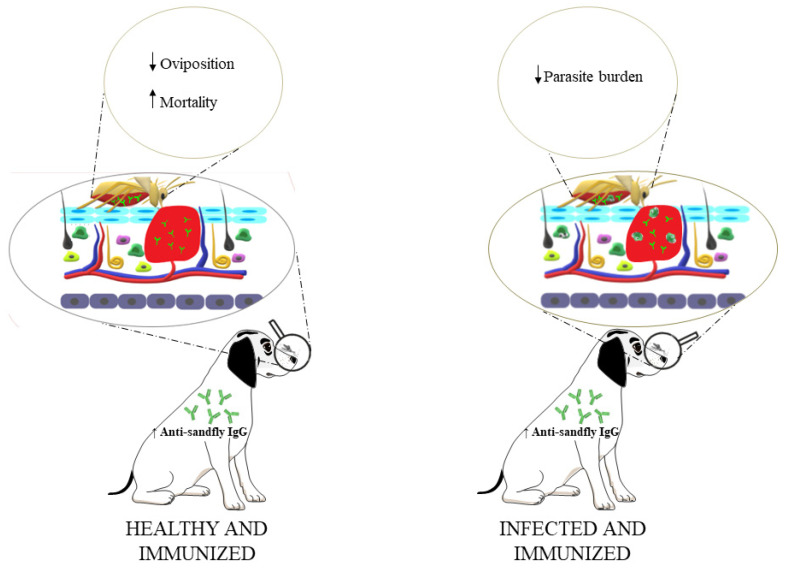
Transmission-blocking vaccines (TBVs) for canine leishmaniasis control. The arrows ↑ or ↓ indicate the increase or decrease in the related parameters evaluated in the sandfly, respectively.

**Table 1 vaccines-11-01565-t001:** The main results of commercial vaccines for CVL.

Continent/ Country	Product	Company	Vaccine Composition	Efficacy Biomarkers	References
Brazil	Leish-Tec^®^	CEVA (France)	Recombinant A2 (rA2) protein with the adjuvant saponin	↑IgG, IgG2, IFN-γ and IL-10; parasites detected in 57.14% (bone marrow culture) and 28.5% (blood PCR) in vaccinated dogs	Fernandes et al., 2008 [47]
				↑IgG, IgG2 and IgG1; 58.1% efficacy (bone marrow culture + xenodiagnoses)	Regina-Silva et al., 2016 [46]
				↑IgG and ↑IgG2; 43% of vaccinated dogs developed the disease	Grimaldi et al., 2017 [48]
				↑ CD8^+^ IFN-γ^+^	Aguiar-Soares et al., 2020 [49]
Europe	LetiFend^®^	LETI Laboratories (Spain)	Protein Q–a genetic fusion of five antigenic fragments from four *L. infantum* proteins, named acidic ribosomal proteins Lip2a, Lip2b, LiP0, and the histone H2A	↑ DTH (9/10 vaccinated dogs); 90% of vaccinated dogs remain healthy (lymph nodes culture, clinical and anatomic-pathologic analysis)	Molano et al., 2003 [51]
				↑ NO production and DTH; parasites detected in vaccinated dogs (single dose): 1/7, 1/7 and 0/7 (PCR of skin, lymph node and spleen, respectively); parasites detected in vaccinated dogs (two doses): 4/7, 1/7 and 2/7 (PCR of skin, lymph node and spleen, respectively)	Carcelén et al., 2009 [52]
				↑IgG2 anti-Protein Q; 72% efficacy (lymph nodes or bone marrow PCR and smear)	Cotrina et al., 2018 [53]

The arrow (↑) indicate the increase in biomarker levels, when compared to control groups. DTH: delayed hypersensitivity; NO: nitric oxide.

**Table 2 vaccines-11-01565-t002:** The main results of TBVs for leishmaniasis.

TBV Composition	Vector	Parasite	Vaccination Schedule/ Animals	Artificial or In Vivo Feeding	Evaluated Parameters	Main Findings	Reference
Sandfly gut antigens	*Phlebotomus duboscqi*	-	1 IM dose, followed by 2 SC doses (14th and 21st day)/24 hamsters	In vivo feeding	Humoral response; Survival and fecundity of sandflies	↑ *P. duboscqi*-specific IgG; ↓ survival; egg production and egg hatching	Ingonga et al., 1996 [70]
Crude whole *Leishmania major* parasites or rgp63 or LPG or rgp63/LPG	*Phlebotomus duboscqi*	*Leishmania major*	4 IV doses, at 7-day interval or 3 IV doses, at 14-day interval/BALB/c mice, posteriorly infected with *L. major*	In vivo feeding	Humoral response; Infection rate in sandfly; Promastigote forms presented after blood meal; Histopathology of midgut	↑ IgG anti-soluble *L. major* antigen; ↓ infection rate; ↓ infective metacyclic forms	Tonui et al., 2001 [71]
PpGalec	*Phlebotomus papatasi*	*Leishmania major*	5 doses/BALB/c	Artificial feeding	Infection rate by ex vivo and in vivo analyses in sandflies	↓ infection rate	Kamhawi et al., 2004 [72]
Repeated sandfly bites	*Lutzomyia longipalpis*	-	Repeated bites of 100–120 females/Rabits	In vivo feeding	Humoral response; sandfly survival and oviposition analysis	↑ IgG anti-sandfly; ↑ mortality; ↓ oviposition	Vilela et al., 2006 [73]
Leishmune^®^	*Lutzomyia longipalpis*	*Leishmania chagasi*	3 SC doses at 20-day interval/mongrel dogs	Artificial feeding	Infection rate (in vitro and in vivo analysis) in sandflies	↑ *L. chagasi* binding to sandfly midguts; ↓ infection rate	Saraiva et al., 2006 [74]
PpChit1	*Phlebotomus papatasi*	*Leishmania major*	3 SC doses at 14-day interval/BALB/c	Artificial feeding	Infection rate in sandflies	↓ infection rate	Coutinho-Abreu et al., 2010 [76]
CaniLeish^®^	*Phlebotomus perniciosus*	*Leishmania infantum*	3 doses/beagle dogs, natural infected after vaccination	In vivo feeding	Infection rate in sandflies	↓ infection rate	Bongiorno et al., 2013 [77]

The arrows (↑ and ↓) indicate the increase and decrease in biomarker levels, respectively, when compared to control groups. IgG: immunoglobulin; IM: intramuscular; IV: intravenous; SC: subcutaneous.

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
