# Peer review of "Transmission-Blocking Vaccines for Canine Visceral Leishmaniasis: New Progress and Yet New Challenges"

_vaccines, 2023, doi:10.3390/vaccines11101565_

Round 1

Reviewer 1 Report

All comments are in the revised PDF

English quality is excellent

Author Response

# I suggest to provide more details for VL before describing the control measures

We are thankful for this comment. We have included additional details, as follows:

Lines 37-41: “Visceral leishmaniasis (VL), the most severe form of leishmaniasis [1, 2] is caused by Leishmania (Leishmania) donovani [3] in the Old World, and Leishmania (Leishmania) infantum [4] in both the Old and New World (syn L. chagasi). VL, caused by L. infantum, is a zoonosis with the dog (Canis familiaris) playing a key role as reservoirs of L. infantum protozoan, once canine host is the main domestic reservoir [5, 6, 7].

# Please replace by a reference number and rearrange the references thereafter

We are thankful for this observation. The reference was replaced by a reference number and also included in the reference section, as follows:

Lines 78-80: “The compartmentalized organ-specific immune response has been associated with both granuloma maturation in liver and antileishmanial activity in L. donovani infection [33].”

Lines 818-820: “33. Stäger S.; Alexander, J.; Carter, K. C.; Brombacher, F.; Kaye, P. M. Both interleukin-4 (IL-4) and IL-4 receptor alpha signaling contribute to the development of hepatic granulomas with optimal antileishmanial activity. Infect Immun 2003, 71, 4804-7, doi.org/10.1128/IAI.71.8.4804-4807.2003. “

# Nitric oxide (NO)

We are thankful for this observation. In fact, the paragraph corresponding to this observation was removed from the text, since the marketing of CaniLeish was suspended in Europe.

# Write the reference No.

We appreciate this comment. However, the paragraph corresponding to this observation was removed from the text, since the marketing of CaniLeish was suspended in Europe.

# Delete the year, it is enough as the reference number is provided

We appreciate this observation. The year was deleted in all similar citations. All changes were performed using the word track changes.

# It could be changed to coauthors, team or coworkers

We are thankful for this comment. We have changed to coworkers.

# Proposed what?

We have changed the sentence, as follows:

Line 242- 243: “Bongiorno et al. (2013) evaluated the ability of the vaccine CaniLeish® to act as a TBV”.

# Please add recent references from 2020-2023

We are thankful for this comment. We have included new references, as follows:

[1] Mann et al., 2021: Mann, S.; Frasca, K.; Scherrer, S.; Henao-Martínez, A.F.; Newman, S.; Ramanan, P.; Suarez, J.A. A Review of Leishmaniasis: Current Knowledge and Future Directions. Curr Trop Med Rep 2021, 8, 121-132, doi.org/10.1007/s40475-021-00232-7.

[7] Scarpini et al., 2022: Scarpini, S.; Dondi, A.; Totaro, C.; Biagi, C.; Melchionda, F.; Zama, D.; Pierantoni, L.; Gennari, M.; Campagna, C.; Prete, A.; Lanari, M. Visceral Leishmaniasis: Epidemiology, Diagnosis, and Treatment Regimens in Different Geographical Areas with a Focus on Pediatrics. Microorganisms 2022, 10, 1887, doi.org/ 10.3390/microorganisms10101887.

[14] WHO, 2023: WHO,2023. Leishmaniasis: Key facts. Available online at: https://www.who.int/news-room/fact-sheets/detail/leishmaniasis. Access in: 09/05/2023.

[45] Baxarias et al. 2022: Baxarias, M.; Homedes, J.; Mateu, C.; Attipa, C.; Solano-Gallego, L. Use of preventive measures and serological screening tools for Leishmania infantum infection in dogs from Europe. Parasit Vectors 2022, 15, 134, doi.org/10.1186/s13071-022-05251-5.

[79] Quintela-Carvalho et al 2022: Quintela-Carvalho, G.; Goicochea, A. M. C.; Mançur-Santos, V.; Viana, S. M.; Luz, Y. D. S.; Dias, B. R. S.; Lázaro-Souza, M.; Suarez, M.; de Oliveira, C. I.; Saraiva, E. M.; Brodskyn, C. I.; Veras, P. T.; de Menezes, J. P. B.; Andrade, B. B.; Lima, J. B.; Descoteaux, A.; Borges,V. M. Leishmania infantum Defective in Lipophosphoglycan Biosynthesis Interferes With Activation of Human Neutrophils. Front Cell Infect Microbiol 2022 12, 788196, doi.org/10.3389/fcimb.2022.788196. 

[80] Cardoso et al. 2020: Baxarias, M.; Homedes, J.; Mateu, C.; Attipa, C.; Solano-Gallego, L. Use of preventive measures and serological screening tools for Leishmania infantum infection in dogs from Europe. Parasit Vectors 2022, 15, 134, doi.org/10.1186/s13071-022-05251-5.

Reviewer 2 Report

The manuscript "Transmission-blocking vaccines for canine visceral leishmaniasis: new
progress and yet new challenges" is very interesting.
The review is well written and
brings the main findings about
transmission-blocking vaccines for canine visceral
leishmaniasis
, which is indeed an interesting strategy in the fight against this neglected
disease.  My main suggestion would be for the authors to report which of these studies
resulted in patent deposits, making a critical analysis of the results found, since this
type of vaccine has great biotechnological potential, and should not only be restricted
to academic studies and it is important that they are focused on the patenting of
products.

Author Response

We are thankful for this comment. In fact, these products have great biotechnological potential and may contribute greatly to the control of leishmaniasis in general. After conducting a search in the main patent databases, using the keywords “transmission block vaccine” or the name of which antigen or the name of inventor, we verified that there are patents only for CaniLeish® and Leishmune®. Moreover, some studies, like those developed by Ingonga et al. 1996 and Vilela et al. 2006, used methodologies or antigens that cannot be patented. In this sense, we included a short discussion about this subject, as follows:

Line 257-260: “Moreover, despite the great biotechnological potential of TBVs, to our knowledge, none of the studies cited above resulted in patents deposit. In fact, only patents related to CaniLeish® and Leishmune® were found in the databases. However, these vaccines were not reported as TBVs.” 

Reviewer 3 Report

This review manuscript on TBVs in leishmaniasis is well designed and well written, key aspects on this field have been addressed and current situation of TBVs has been thoroughly assessed. I do only have few minor comments:

LINE 67: this reference (Stäger et al., 2003) is not included in the reference section. Also in line 572: two papers are together in reference 75: Secundino et al., (2010) and Petitdidier et al (2019)

LINE 83-85: Although CaniLeish® is authorised for use in the European Union, in Spain, CaniLeish® current situation is similar to that of LEISH-Tec® in Brazil. The Spanish competent authorities have suspended commercialization of the vaccine, please do check the following links:

https://cimavet.aemps.es/cimavet/publico/lista.html#

https://cimavet.aemps.es/cimavet/publico/detalle.html?nregistro=EU/2/11/121/001

https://www.ema.europa.eu/en/medicines/veterinary/EPAR/canileish

I suggest authors to check and address CaniLeish situation in European countries (i.e.: France ¿?)

Along the manuscript, authors addressed different studies, providing a variety of promissing results. Occasionally, these results were presented as statistically significant (line 185, 188, 225 etc) whereas others such as lowest infetion rates (line 2019), parasite burden reduction inside vectors (206) , increased mortality of vectors (line 209) or decrease in the infection index (line 216) were not. I suggest authors not to omit  whether these promising findings were (or were not) statistically significant.

Line 256: typo error ¿?

Author Response

# This review manuscript on TBVs in leishmaniasis is well designed and well written, key aspects on this field have been addressed and current situation of TBVs has been thoroughly assessed. I do only have few minor comments:

# LINE 67: this reference (Stäger et al., 2003) is not included in the reference section. Also in line 572: two papers are together in reference 75: Secundino et al., (2010) and Petitdidier et al (2019)

We are thankful for this observation. The reference was included in the reference section and reference 75 has been corrected, as follows:  

Lines 1021-1023: “78. Secundino, N.; Kimblin, N.; Peters, N. C.; Lawyer, P.; Capul, A. A.; Beverley, S. M.; Turco, S. J.; Sacks, D. Proteophosphoglycan confers resistance of Leishmania major to midgut digestive enzymes induced by blood feeding in vector sand flies. Cellular Microbiology, 2010, 12, 7, 906–918.”

Lines 818-820: “33. Stäger S.; Alexander, J.; Carter, K. C.; Brombacher, F.; Kaye, P. M. Both interleukin-4 (IL-4) and IL-4 receptor alpha signaling contribute to the development of hepatic granulomas with optimal antileishmanial activity. Infect Immun 2003, 71, 4804-7, doi.org/10.1128/IAI.71.8.4804-4807.2003.” 

# LINE 83-85: Although CaniLeish® is authorised for use in the European Union, in Spain, CaniLeish® current situation is similar to that of LEISH-Tec® in Brazil. The Spanish competent authorities have suspended commercialization of the vaccine, please do check the following links:

https://cimavet.aemps.es/cimavet/publico/lista.html#

https://cimavet.aemps.es/cimavet/publico/detalle.html?nregistro=EU/2/11/121/001

https://www.ema.europa.eu/en/medicines/veterinary/EPAR/canileish

I suggest authors to check and address CaniLeish situation in European countries (i.e.: France ¿?)

We are thankful for this comment. We have included the missing information that CaniLeish vaccine was withdrawn from the European market, as follows:

Lines 99-100: “Moreover, after ten years of commercialization, CaniLeish® (Virbac, France) was withdrawn from the European market in 2021 (Baxarias et al. 2022).”

Furthermore, we have removed all CaniLeish information in this section.

# Along the manuscript, authors addressed different studies, providing a variety of promissing results. Occasionally, these results were presented as statistically significant (line 185, 188, 225 etc) whereas others such as lowest infetion rates (line 2019), parasite burden reduction inside vectors (206), increased mortality of vectors (line 209) or decrease in the infection index (line 216) were not. I suggest authors not to omit  whether these promising findings were (or were not) statistically significant.

We are thankful for this comment. In fact, the text was giving gaps to understand that some results cited here were statistically different and others were not. Therefore, the text was adapted so as not to lead the reader to understand that some of the results cited here are not statistically different. All changes were performed using the word track changes.

# Line 256: typo error ¿?

We thank Reviewer #3 for the critical reading and contributing to further improve the quality of the manuscript. We have made suitable changes throughout the manuscript as suggested. We have removed all misspellings and made appropriate changes in the revised version of manuscript.

Reviewer 4 Report

The review by Leite was intended to review the topic related to transmission-blocking vaccines (TBVs) in canine leishmaniasis. This work lacks novelty considering a lot of reviews on "vaccines VS canine leishmaniasis".  In addition, this work explored a very common subject in leishmaniasis, that is immunity: again, more detailed articles on it can be found in literature.

Furthermore, there is only one page on TBVs that would be the main focus of this work, and unfortunately does not bring updated information on the field.

There are a few spelling errors.

Author Response

# The review by Leite was intended to review the topic related to transmission-blocking vaccines (TBVs) in canine leishmaniasis. This work lacks novelty considering a lot of reviews on "vaccines VS canine leishmaniasis".  In addition, this work explored a very common subject in leishmaniasis, that is immunity: again, more detailed articles on it can be found in literature.

Furthermore, there is only one page on TBVs that would be the main focus of this work, and unfortunately does not bring updated information on the field.

We appreciate this comment. In fact, there are several in-depth reviews in the literature about immunology of canine visceral leishmaniasis. However, for better contextualization, and thus provide a better reading, we brought a brief topic on immunology and commercial vaccines for canine visceral leishmaniasis. However, the main objective of the manuscript is to describe all TBV studies for canine visceral leishmaniasis. After exploring all the literature, we showed that, unfortunately, there are still very few studies in this field. In fact, this shows a major bottleneck in scientific research, since TBVs can be great allies in controlling parasite transmission. In this sense, we hope that this review on TBVs, short due to the few studies published so far, will have a positive impact on the rationale for new studies to control the transmission of the parasite, since the few studies published so far have shown promising results. Taking in account that the identification of TVBs with high vaccinal performance against leishmaniasis is extremally important and, despite the few studies in this field, this approach should be considered as an important alternative and complementary intervention for disease control.

Reviewer 5 Report

This is a review article that summarizes the past and present research for developing an effective transmission blocking vaccine (TBV) for canine Leishmaniasis. Dogs are known reservoirs of Leishmania infantum, that causes visceral leishmaniasis in humans. There are a few canine Leismania vaccines those are moderately effective. However, it is now necessary to develop more effective TBV to control the spread of this fatal disease. Therefore, this review is appropriate in time to provide an up-to-date information regarding this matter. 

The topic is relevant in current situation of Leismania vaccine research. 

Tables are very helpful.

Table 2 is little difficult to follow. It would be better to make separate columns for vector and parasite. Similarly, separate columns for vaccination method and vector feeding would be better. 

Author Response

# This is a review article that summarizes the past and present research for developing an effective transmission blocking vaccine (TBV) for canine Leishmaniasis. Dogs are known reservoirs of Leishmania infantum, that causes visceral leishmaniasis in humans. There are a few canine Leismania vaccines those are moderately effective. However, it is now necessary to develop more effective TBV to control the spread of this fatal disease. Therefore, this review is appropriate in time to provide an up-to-date information regarding this matter. 

The topic is relevant in current situation of Leismania vaccine research. 

Tables are very helpful.

Table 2 is little difficult to follow. It would be better to make separate columns for vector and parasite. Similarly, separate columns for vaccination method and vector feeding would be better. 

We thank the Reviewer #5 for the comments regarding our manuscript. For a better comprehension we adapted the columns in the Table 2 as suggested.

Reviewer 6 Report

In this review Leite et al., present the current protective vaccines that are available against CVL, CaniLeish®, Leish-tec® and LetiFend®and underline their efficacy and their limited effectiveness. The authors highlight the need to explore other routes for protection against leishmania trasmission. Hence they focus on presenting the strategy of  transmission-blocking vaccines (TBVs). They exhibit the current research efforts on TBVs against leishmania trasmission, their advantages but also the disadvantages and challenges of this strategy. This mini review can be published in vaccines journal after minor changes.

Line 67: Please insert the correct format of the reference

Line 84-85: Please elaborate shortly why they suspended the manufacture and sale of Leish-Tec

Line 94: "...an in vivo increase of IFN-γ and NO production while the efficacy of vaccination was 92%"

Line 102: "...supplemented with..."

Line 103-104: Please rephrase the sentence to be more clear and comprehensible

Line 110: use the word "with" instead of "than"

Line 123-124: Please rephrase the sentence to be more clear and comprehensible

Line 131: use the word "get" instead of "be"

Line 131-132: "Some researchers consider their efficacy limited and believe that vaccines may interfere with the interpretation of serological tests for dog disease diagnosis"

Line 191: mention some of the rest diseases

Line 214-215: Please rephrase the sentence to be more clear and comprehensible

Line 216: use another word instead of demonstrated

Line 217-221: Please rephrase the sentence to be more clear and comprehensible

Author Response

# In this review Leite et al., present the current protective vaccines that are available against CVL, CaniLeish®, Leish-tec® and LetiFend®and underline their efficacy and their limited effectiveness. The authors highlight the need to explore other routes for protection against leishmania trasmission. Hence they focus on presenting the strategy of  transmission-blocking vaccines (TBVs). They exhibit the current research efforts on TBVs against leishmania trasmission, their advantages but also the disadvantages and challenges of this strategy. This mini review can be published in vaccines journal after minor changes.

# Line 67: Please insert the correct format of the reference

We are thankful for this observation. The correct reference format was included, as follows:

Line 78-80: “The compartmentalized organ-specific immune response has been associated with both granuloma maturation in liver and antileishmanial activity in L. donovani infection [33].”

#Line 84-85: Please elaborate shortly why they suspended the manufacture and sale of Leish-Tec

We appreciate this comment. A short explanation was included, as follows:

Lines 96-99: “However, more recently, after verifying product compliance deviation, which may compromise the vaccine's effectiveness, Brazilian Ministry of Agriculture and Livestock determined the suspension of the manufacture and sale of Leish-Tec® and, so far, new recommendations are being awaited [44].”

#Line 94: "...an in vivo increase of IFN-γ and NO production while the efficacy of vaccination was 92%"

In fact, this paragraph was removed from the text, since the marketing of CaniLeish was suspended in Europe.

#Line 102: "...supplemented with..."

We are thankful for this comment. As suggested, the sentence in question can be found in the text as follows:

Lines 108-110: “It consists of recombinant A2 (rA2) protein, which was the first to be identified as the amastigote-specific virulence factor in Leishmania [46], supplemented with the adjuvant saponin.”

#Line 103-104: Please rephrase the sentence to be more clear and comprehensible

The sentence was reformulated, as follows:

Lines 105-106: “However, despite the high IFN- γ production, parasites were detected in 57.14% and 28.5% of the bone marrow and blood of vaccinated animals, respectively.”

#Line 110: use the word "with" instead of "than"

The word “than” has changed, as follows:

Lines 195-196:Subsequently, Grimaldi et al. showed the same antibody production with the aforementioned study in vaccinated animals in a field trial.”

#Line 123-124: Please rephrase the sentence to be more clear and comprehensible

We are thankful for this comment. For a better comprehension, the sentence was reformulated, as follows:

Lines 208-210: “Moreover, the vaccine triggered a positive delayed-type hypersensitivity (DTH) and, after experimental infection, vaccinated dogs showed lower intensity of clinical symptoms”.

#Line 131: use the word "get" instead of "be"

We appreaciate this comment. The word “be” was changed, as follows:

Line 214-216: “The studies briefly discussed above indicate that the commercially available vaccines have limitations and none of them are 100% effective, meaning that vaccinated dogs could still get infected with L. infantum.”

#Line 131-132: "Some researchers consider their efficacy limited and believe that vaccines may interfere with the interpretation of serological tests for dog disease diagnosis"

We are thankful for this comment. As suggested, the revised can be found in the text as follows:

Lines 216-217: “Some researchers consider their efficacy limited and believe that vaccines may interfere with the interpretation of serological tests for dog disease diagnosis [13].”

#Line 191: mention some of the rest diseases

We are thankful for this comment. The sentence was reformulated, as follows:

Line 199-201: “Finally, TBVs have already been described for other diseases and vector, such as West Nile virus and ticks [65].”

#Line 214-215: Please rephrase the sentence to be more clear and comprehensible

We are thankful for this comment. For a better comprehension, the sentence was reformulated, as follows:

Lines 491-494: “After animals’ vaccination, an artificial sandfly feeding was employed in order to access the infection of L. longipalpis females with L. chagasi. For this purpose, a chick-skin membrane was used, where sandfly fed of dog’s immunized sera plus L. chagasi parasites.”

# Line 216: use another word instead of demonstrated

The word “demonstrated” was changed, as follows:

Lines 493-495: “The authors observed a decrease of 20.7% in infection index, proved by the ability to block the attachment of parasites in the midgut and by the lower rate of parasite infection [70].”

# Line 217-221: Please rephrase the sentence to be more clear and comprehensible

We appreciate this comment. For a better comprehension, the sentence was reformulated, as follows:

Lines 497-500: “Although Leishmune® did not contain vector antigens in its formulation this vaccine was able to interfere in Leishmania infection in sandflies.. However, in 2014 its commercialization was suspended in Brazil for not meeting the requirements of the phase III study [71].”